# Comparison of Oncologic Outcomes between Transduodenal Ampullectomy and Pancreatoduodenectomy in Ampulla of Vater Cancer: Korean Multicenter Study

**DOI:** 10.3390/cancers13092038

**Published:** 2021-04-23

**Authors:** Seung-Soo Hong, Sung-Sik Han, Wooil Kwon, Jin-Young Jang, Hee-Joon Kim, Chol-Kyoon Cho, Keun-Soo Ahn, Jae-Do Yang, Youngmok Park, Seog-Ki Min, Ju-Ik Moon, Young-Hoon Roh, Seung-Eun Lee, Joon-Seong Park, Sang-Geol Kim, Chi-Young Jeong, Jin-Seok Heo, Ho-Kyoung Hwang

**Affiliations:** 1Department of Hepatobiliary and Pancreatic Surgery, Yonsei University College of Medicine, Seoul 03722, Korea; hongss@yuhs.ac; 2Department of Surgery, National Cancer Center, Goyang 10408, Korea; sshan@ncc.re.kr; 3Department of Surgery, Seoul National University College of Medicine, Seoul 03080, Korea; willdoc@snu.ac.kr (W.K.); jangjy4@snu.ac.kr (J.-Y.J.); 4Department of Surgery, Chonnam National University Medical School, Gwangju 61469, Korea; heejoonkim@jnu.ac.kr (H.J.K.); ckcho@jnu.ac.kr (C.K.C.); 5Department of Surgery, Keimyung University Dongsan Hospital, Keimyung University School of Medicine, Daegu 42601, Korea; ahnks@dsmc.or.kr; 6Department of Surgery, Chonbuk National University College of Medicine, Jeonju 54907, Korea; hirojawa@jbnu.ac.kr; 7Division of HBP Surgery, Department of Surgery, Biomedical Research Institute, Pusan National University Hospital, Gudeok-ro, Seo-gu, Busan 49241, Korea; pym777@pnuh.co.kr; 8Department of Surgery, Ewha Womans University College of Medicine, Seoul 07985, Korea; mp9666@ewha.ac.kr; 9Department of Surgery, Konyang University Hospital, Deajeon 35365, Korea; monjuik@kyuh.ac.kr; 10Department of Surgery, Dong-A University College of Medicine, Busan 49201, Korea; gsryh@dau.ac.kr; 11Department of Surgery, Chung-Ang University Hospital, Chung-Ang University, College of Medicine, Seoul 06973, Korea; selee508@cau.ac.kr; 12Department of Surgery, Gangnam Severance Hospital, Yonsei University College of Medicine, Seoul 06273, Korea; JSPARK330@yuhs.ac; 13Department of Surgery, Kyungpook National University College of Medicine, Daegu 41944, Korea; ksg@knu.ac.kr; 14Department of Surgery, Gyeongsang National University School of Medicine, Jinju 52727, Korea; drjcy@gnu.ac.kr; 15Division of Hepatobiliary-Pancreatic Surgery, Department of Surgery, Samsung Medical Center, Sungkyunkwan University School of Medicine, 81, Ilwon-Dong, Gangnam-gu, Seoul 06351, Korea

**Keywords:** ampulla of Vater cancer, transduodenal ampullectomy, pancreaticoduodenectomy

## Abstract

**Simple Summary:**

This study used multicenter data to compare the oncological safety of transduodenal ampullectomy (TDA) with that of pylorus-preserving pancreatoduodenectomy (PPPD) in early ampulla of Vater (AoV) cancer. Data for patients who underwent surgical resection for AoV cancer (pTis–T2 stage) from 2000 to 2019 were collected from 15 institutions. A total of 486 patients were enrolled (PPPD, 418; TDA, 68). The oncologic behavior (tumor size, T stage, differentiation, lymphovascular invasion) in the PPPD group was more aggressive than that in the TDA group at all T stages. The 5-year disease-free survival and overall survival did not differ between the two groups when considering all T stages or only the Tis + T1 group. In T1 patients, PPPD had survival outcomes superior to those in the TDA group. In the TDA group, lymph node dissection did not affect survival. In conclusion, PPPD should be the standard procedure for early AoV cancer.

**Abstract:**

This study used multicenter data to compare the oncological safety of transduodenal ampullectomy (TDA) with that of pylorus-preserving pancreatoduodenectomy (PPPD) in early ampulla of Vater (AoV) cancer. Data for patients who underwent surgical resection for AoV cancer (pTis–T2 stage) from January 2000 to September 2019 were collected from 15 institutions. The clinicopathologic characteristics and survival outcomes were compared between the PPPD and TDA groups. A total of 486 patients were enrolled (PPPD, 418; TDA, 68). The oncologic behavior in the PPPD group was more aggressive than that in the TDA group at all T stages: larger tumor size (*p* = 0.034), advanced T stage (*p* < 0.001), aggressive cell differentiation (*p* < 0.001), and more lymphovascular invasion (*p* = 0.002). Five-year disease-free survival (DFS) and overall survival (OS) did not differ between the two groups when considering all T stages or only the Tis+T1 group. Among T1 patients, PPPD produced significantly better DFS (PPPD vs. TDA, 84.8% vs. 66.6%, *p* = 0.040) and superior OS (PPPD vs. TDA, 89.1% vs. 68.0%, *p* = 0.056) than TDA. Lymph node dissection (LND) in the TDA group did not affect DFS or OS (TDA + LND vs. TDA-only, DFS, *p* = 0.784; OS, *p* = 0.870). In conclusion, PPPD should be the standard procedure for early AoV cancer.

## 1. Introduction

Ampulla of Vater (AoV) cancer is a malignant neoplasm that develops from the ampulla of Vater complex, which lies between the area distal to the confluence of the pancreatic duct and the distal common bile duct (CBD) and the duodenal opening. AoV cancer is a rare disease that constitutes roughly 7% of peri-ampullary tumors [1,2]. Although AoV cancer is a peri-ampullary tumor, it has a better prognosis than other peri-ampullary tumors such as pancreatic cancer and distal CBD cancer [3,4,5]. Approximately half of patients with AoV cancer present at an advanced stage. For those who present at an early stage, pancreaticoduodenectomy (PD) or pylorus-preserving pancreaticoduodenectomy (PPPD) are recognized as standard treatments [6,7,8]. Because the disease generally presents in patients whose age makes surgical procedures risky, only 40% of AoV patients undergo surgical resection [9].

Transduodenal ampullectomy (TDA) has lower peri-operative morbidity and mortality than PD and has been suggested as an alternative surgical treatment for ampullary adenoma and early AoV cancer [10,11,12]. However, TDA presents a higher chance of recurrence due to limited resection and insufficient lymph node dissection, and its oncological safety has been controversial [13]. Therefore, TDA is used only in patients with high surgical co-morbidity factors, small cancer with good cell differentiation, and staging equivalent to or less than T1. The indications for TDA differ among institutes, and no clear guidelines have been established [14,15]. Some previous studies have compared the oncologic outcomes of PD and TDA in early AoV cancer. Winter et al. from the Johns Hopkins group reported no statistical differences (*p* = 0.150) in postoperative complications between PD (*n* = 435) and TDA (*n* = 15). They noticed a high incidence of lymph node metastasis (28.0%) in T1 cancer and concluded that PD should be preferred, even in early AoV cancer [16]. A retrospective single-institution study of early AoV cancer in China (PD: *n* = 21, TDA: *n* = 22) reported no statistical differences between the two groups in 5-year survival outcomes and a lower complication rate in the TDA group [17]. A retrospective single-institution study in Korea evaluated the clinicopathologic characteristics, disease-free survival (DFS), and recurrence rate in 137 patients (PD: *n* = 119, TDA: *n* = 18) with early AoV cancer (Tis or T1). Those authors reported no statistical differences in postoperative complications or DFS between the two groups; however, among patients with T1 cancer, the TDA group showed a statistically higher recurrence rate than the PD group [18]. The results of those studies have limited reliability due to the limitations of single-institution research and the relatively small number of patients who underwent TDA. Therefore, we conducted a large multicenter study in Korea to evaluate the oncological safety of TDA in early AoV cancer. We compared the clinicopathologic characteristics, DFS, and overall survival (OS) of early AoV cancer patients treated with PD or TDA and sought applicable indications for TDA.

## 2. Materials and Methods

### 2.1. Patient Selection and Evaluation

This retrospective multicenter cohort study began with data from 2278 patients registered in the Korean Tumor Registry System Biliary and Pancreas (KOTUS-BP) who were diagnosed with AoV adenoma or carcinoma in any of 24 hospitals in Korea from January 2000 to September 2019. The raw data included 15 cases of total pancreatectomy, 2123 cases of PPPD, 125 cases of TDA, 22 cases of biopsy only, and 23 cases of bypass surgery. Among those patients, 418 who had pathologic Tis, T1, or T2 disease without distant metastasis, combined organs, or a vascular resection were included in the PPPD group from 15 institutions. Even though pT2 is not early AoV cancer, 13 patients in the KOTUS-BP data underwent TDA for T2 cancer, so we included them in this study to analyze their characteristics and oncologic outcomes. The TDA group contained 68 patients who underwent TDA to treat pathologic Tis, T1, or T2 disease (Figure 1).

We compared the clinicopathologic characteristics (age, sex, tumor size, T/N staging, tumor differentiation, lymphovascular invasion (LVI), perineural invasion (PNI), adjuvant treatment, and R status of operation) between the TDA and PPPD groups. T and N staging were based on the American Joint Committee on Cancer (AJCC) 7th edition. When analyzing the recurrence patterns, local recurrence was defined as tumor recurrence or lymph node metastasis around the surgical site, and systemic recurrence was defined as metastasis to distant organs or lymph nodes beyond the surgical field or peritoneal dissemination. DFS and OS were compared between the TDA and PPPD groups, and the TDA group was subdivided into the TDA with lymph node dissection (LND, TDA + LND) group and the TDA without LND (TDA-only) group to compare survival differences according to LND status. To reduce heterogeneity and set appropriate indications for TDA, we conducted a subgroup analysis of the Tis and T1 group and the T1 only group.

In addition, because in-depth information about the decision to perform PPPD or TDA was unavailable in these data, which we collected retrospectively from multiple centers, we additionally conducted a propensity score matching (PSM) analysis and compared clinicopathologic characteristics and survival outcomes between those groups to reduce selection bias.

In univariate and multivariate analyses, we investigated the risk factors that affected DFS and OS. Additionally, the ratio of lymph node metastasis according to T-stage was investigated but is not included in the tables. This study was approved by the Institutional Review Board of Yonsei University College of Medicine (4-2020-1144).

### 2.2. Statistical Analysis

All statistical analyses were performed using the Statistical Package for Social Sciences version 23 (SPSS, Chicago, IL, USA). Values are expressed as means and standard deviations or medians and ranges, as appropriate. Categorical variables were compared using the Chi-square test and are reported as number (n) and percentage (%). Continuous variables were compared using independent t-testing or Mann–Whitney testing, as appropriate. We conducted a propensity score 1:1 matching analysis by calculating a propensity score for each surgical method using T staging, N staging, cell differentiation, LVI, PNI, adjuvant treatment, and margin status. The *p*-value for statistical significance was set at 0.05. In evaluating the risk factors affecting DFS and OS, we used a Cox proportional hazard model, and variables whose *p*-value was less than 0.05 in the univariate analysis were used in the multivariate analysis.

## 3. Results

### 3.1. Clinicopathologic Characteristics of the PPPD and TDA Groups according to T Stage

Table 1 compares the outcomes and clinicopathologic characteristics of the PPPD and TDA groups according to T staging. Among all patients (Tis, T1, and T2), age and sex did not differ between the two treatment groups, but all the characteristics that reflect pathological severity were significantly more severe in the PPPD group. Tumor size was significantly larger in the PPPD group than in the TDA group (1.91 ± 1.02 vs. 1.62 ± 0.79 cm, *p* = 0.034). Stage pT2 was most common stage in the PPPD group (224 patients, 53.6%), and pT1 was the most common stage in the TDA group (31 patients, 45.6%). For Tis tumors, only 6 patients (1.4%) were included in the PPPD group, whereas 24 patients (35.3%) were included in the TDA group (*p* < 0.001). The N1 stage comprised 25.1% (*n* = 105) of the PPPD group, whereas only 2.9% (*n* = 2) of the TDA group had N1 stage disease. In the TDA group, 42 patients (61.8%) did not undergo LND (*p* < 0.001). When we investigated the ratio of lymph node metastasis according to the T staging of all patients, 35.9% of T2 patients, 10.0% of T1 patients, and none of the Tis patients had lymph node metastasis. The cell differentiation results were significantly more severe in the PPPD group than in the TDA group (*p* < 0.001), with a higher ratio of LVI (*p* = 0.002). Due to the limited resection range of TDA, the TDA group had 6 patients (8.8%) with microscopically residual cancer, whereas only 1 patient (0.2%) had R1 status in the PPPD group (*p* < 0.001). When we analyzed the clinicopathologic findings in early AoV cancer (Tis and T1 patients), the age, sex, tumor size, LVI, PNI, and adjuvant treatment did not differ statistically between the TDA and PPPD groups. However, T staging (*p* < 0.001), N staging (*p* < 0.001), and cell differentiation (*p* < 0.001) were still more severe in the PPPD group than in the TDA group. The R1 rate was higher in the TDA group (0% vs. 5.5%, *p* = 0.010). These differences were similar when we compared only T1 stage patients (Table 1).

### 3.2. Clinicopathologic Characteristics of the PPPD, TDA+LND, and TDA-Only Groups with T1 Stage Disease

Because patients with T1 disease had better DFS following PPPD than TDA (84.8% vs. 66.6%, *p* = 0.040), we analyzed differences in the clinicopathologic characteristics of the T1 stage patients in both groups. For this analysis, we divided surgical methods into three groups: PPPD, TDA+LND, and TDA-only. The PPPD and TDA+LND groups did not differ in age, sex, tumor size, N stage, cell differentiation, LVI, PNI, or adjuvant treatment. However, the R0 resection rate was higher in the PPPD group than in the TDA+LND group (100% vs. 84.6%, *p* = 0.004). In the comparison between the PPPD and TDA-only groups, the tumors were significantly larger (1.8 vs. 1.2 cm, *p* = 0.025), and cell differentiation was worse in the PPPD group than in the TDA-only group (*p* = 0.008) (Table 2).

### 3.3. Comparison of Survival Outcomes

We analyzed the 5-year DFS and OS rates by dividing the study population into three groups by T stage, and we compared the PPPD and TDA groups using the entire population (Tis, T1, and T2), Tis + T1, and T1-only. We also performed survival analysis by dividing the TDA group into TDA+LND and TDA-only groups. In the entire study population (Tis, T1, and T2), the overall recurrence rate did not differ between the PPPD and TDA groups (*p* = 0.258). However, the local recurrence rate was higher in the TDA group than in the PPPD group (10.3% vs. 5.5%, *p* = 0.012), and the systemic recurrence rate was higher in the PPPD group (16.7% vs. 5.9%, *p* = 0.012). Within the systemic recurrences, peritoneal seeding was observed in 7 patients (1.7%) in the PPPD group and 1 patient (1.5%) in the TDA group, which was without statistical difference (*p* = 1.000) (Table 1). The 5-year DFS and OS rates did not differ between the PPPD and TDA groups (71.9% vs. 78.6%, *p* = 0.156, 74.2% vs. 77.6%, *p* = 0.816, respectively, Figure 2A,B). When survival outcomes were analyzed among the PPPD, TDA+LND, and TDA-only groups, neither DFS (PPPD vs. TDA+LND vs. TDA-only, 71.9% vs. 79.3% vs. 78.1%, *p* = 0.366) nor OS (74.2% vs. 82.6% vs. 75.6%, *p* = 0.932) differed statistically (Figure 2C,D).

In the Tis + T1 study population, 5-year DFS and OS did not differ statistically between the PPPD and TDA groups (84.5% vs. 80.1%, *p* = 0.596, 88.4% vs. 76.0%, *p* = 0.122, respectively, Figure 3A,B), though local recurrence was more frequently observed in the TDA group (7.3% vs. 1.0%, *p* = 0.033) (Table 1). Even when comparing the PPPD, TDA+LND and TDA-only groups, DFS (*p* = 0.868) and OS (*p* = 0.286) did not differ (Figure 3C,D).

In the T1 population, 5-year DFS was significantly better in the PPPD group than in the TDA group (84.8% vs. 66.6%, *p* = 0.040), and 5-year OS was marginally better in the PPPD group than the TDA group (89.1% vs. 68.0%, *p* = 0.056) (Figure 4A,B). When the TDA group was divided into TDA+LND and TDA-only groups, the PPPD group showed a better 5-year DFS rate than the TDA-only group (84.8% vs. 64.1%, *p* = 0.046). Although there was no statistical significance, 5-year DFS in the PPPD group was also better than that in the TDA+LND group (84.8% vs. 68.8%, *p* = 0.241). Five-year OS did not differ significantly among the three groups (PPPD vs. TDA+LND, 89.1% vs. 69.2%, *p* = 0.138; PPPD vs. TDA-only, 89.1% vs. 67.1%, *p* = 0.138; TDA+LND vs. TDA-only, 69.2% vs. 67.1%, *p* = 0.870); however, the PPPD group had a better prognosis by more than 20% compared with the TDA+LND and TDA-only groups. (Figure 4C,D). When the recurrence pattern was analyzed, local recurrence occurred significantly more often in the TDA-only group than in the PPPD group (16.7% vs. 0.5%, *p* = 0.016) (Table 2).

### 3.4. Propensity Score Matching Analysis

We additionally conducted a PSM analysis and compared the clinicopathologic characteristics and survival outcomes between groups to reduce selection bias. However, even in the PSM analysis, the differences between T staging and N staging did not diminish in the total patient groups (Tis–T2) or the Tis and T1 groups (Appendix A). Among the T1 patients in the PSM analysis, the N stage, cell differentiation, LVI, PNI, adjuvant treatment, and R0 resection rate did not differ between the groups, and the 5-year DFS and OS did not differ between the groups either (DFS, PPPD vs. TDA, 83.20% vs. 66.60%, *p* = 0.125; OS, 89.80% vs. 68.00%, *p* = 0.266).

### 3.5. Analysis of Risk Factors That Affect Survival Outcomes

We evaluated risk factors that affected DFS and OS across the entire study population (Table 3). In the univariate analysis, being 65 years or older (OR: 1.502, *p* = 0.038), pT2 (OR: 10.869, *p* = 0.018), pN1 (OR: 3.841, *p* < 0.001), or having aggressive cell differentiation (moderate, OR: 2.574, *p* < 0.001; poorly differentiated, OR: 4.002, *p* < 0.001), LVI (OR: 2.580, *p* < 0.001), or PNI (OR: 1.621, *p* = 0.038) were significant risk factors affecting DFS. In the multivariate analysis, pN1 staging (OR: 2.030, *p* = 0.003) and moderate cell differentiation (OR: 1.778, *p* = 0.041) remained significant risk factors for poor DFS. In the analysis of risk factors for OS, pN1, moderate differentiation, and LVI were significant risk factors in both the univariate and multivariate analyses. pN1 and moderate differentiation were thus identified as significant risk factors affecting both DFS and OS in the multivariate analyses. The operation methods did not affect DFS or OS in the univariate or multivariate analyses.

Similar results were obtained in the risk factor analysis for the T1 stage group. pNx (OR: 3.045, *p* = 0.024), pN1 (OR: 7.920, *p*< 0.001), advanced cell differentiation (moderate: OR 3.035, *p* = 0.012; poorly: OR 11.104, *p* < 0.001), LVI (OR: 4.108, *p* = 0.003), and PNI (OR: 4.124, *p* = 0.002) were significant risk factors affecting DFS in the univariate analysis. In the multivariate analysis, poor differentiation (OR: 5.446, *p* = 0.041) and PNI (OR: 2.953, *p* = 0.047) remained significant risk factors for poor DFS. In the analysis of risk factors affecting OS, pN1 (OR: 3.898, *p* = 0.046) and LVI (OR: 6.933, *p* = 0.010) were significant risk factors in the multivariate analysis (Table 4). The operation methods did not affect DFS or OS in the univariate or multivariate analyses.

## 4. Discussion

The first local resection of an ampullary cancer was reported by Halsted in 1899 under the name TDA [19]. Walter Kausch first attempted a resection of the majority of the duodenum en bloc with a significant portion of the pancreas in 1912 [20]. The radical resection of peri-ampullary cancer went through a two-stage operation period [21,22,23], and in 1940, A.O. Whipple reported the modern concept of PD, including complete excision of the head of the pancreas and the entire duodenum [24]. TDA has failed to become widespread because the surgical technique has not been standardized, and apprehension about the possibility of recurrence has persisted [19]. Currently, PD, the standard surgical strategy for ampullary cancer, is still associated with a high rate (33%–52%) of postoperative complications [16,25,26]. Therefore, TDA is applied in patients who are not candidates for PD due to comorbidities [14,15]. The appropriate indications for TDA remain controversial. Some studies [16,17,18,25] have compared the oncological outcomes of PD and TDA, but they failed to provide conclusive indications for TDA due to a small number of cases and patient heterogeneity.

To secure a sufficient number of cases and solve the problem of heterogeneity, we conducted a multicenter study with data from 15 tertiary hospitals in Korea and compared the oncologic outcomes of PPPD and TDA patients according to T stage. In the comparison of all T stages (Tis, T1, and T2), the DFS and OS did not differ statistically between the PPPD and TDA groups, even though the PPPD group had more advanced disease: larger tumor size, more T2 stage patients, more N1 stage patients, and aggressive cell differentiation (Table 1). Considering those clinicopathologic characteristics, the survival outcomes of the TDA group would have been expected to be better, but the results showed no differences. When the survival outcomes were analyzed by dividing the patients into three groups according to the surgical method, PPPD, TDA+LND, and TDA-only, we again found no differences in survival outcomes among the groups (5-year DFS, *p* = 0.366; 5-year OS, *p* = 0.932, Figure 2C,D). These results indicate that TDA (with or without LND) might not be an appropriate surgical method for the treatment of AoV cancer, including T2 cases, compared to PPPD. In addition, because the lymph node metastasis rate at the T2 stage in this study was quite high (35.9%), TDA alone should be avoided in patients at the T2 stage. Lai et al. reported the oncological outcomes of TDA and PD in patients with AoV cancer, including T2 or higher. They compared 362 patients who underwent PD with 15 TDA patients. Of the 15 patients in the TDA group, 7 patients had benign adenoma, 3 patients had Tis, 1 had T1 disease, and 4 had T2 disease; the PD group included patients at all T stages from benign adenoma to T3 or higher. In the pTis, T1, and T2 patients, tumor recurrence occurred in 2 patients (25.0%) in the TDA group and 42 patients (16.6%) in the PD group, a statistically insignificant difference. The T stages of the 2 TDA patients with recurrence were Tis and T2, and they suffered progression to the liver and carcinomatosis, respectively. Even though those authors found no difference in DFS and OS between the TDA and PD groups, as we did here, they concluded that PD is preferable for treating AoV cancer due to the inaccuracy of preoperative biopsy results and the risk of lymph node metastasis [27].

We also compared the oncological outcomes of patients with early AoV cancer (Tis and T1), excluding those with T2 stage disease, and we found results similar to those from our analysis of all patients (Tis–T2). Even though the PPPD group had more aggressive clinicopathologic characteristics (Table 1), the survival rates between the PPPD and TDA groups did not differ (Figure 3). Local recurrence occurred more frequently in the TDA group (7.3% vs. 1.0%, *p* = 0.033, Table 1), but the total recurrence rate did not differ between the groups.

In the survival analysis of T1 patients, the DFS of the PPPD group was significantly superior to that of the TDA group, by about 20% (PPPD vs. TDA, 84.8% vs. 66.6%, *p* = 0.040). The OS of the PPPD group was also better than that of the TDA group by about 20%, which had marginal significance (PPPD vs. TDA, 89.1% vs. 68.0%, *p* = 0.056). Furthermore, in the subgroup analysis in which we divided the TDA group into TDA + LND and TDA-only groups, although statistical significance was found only in DFS between the PPPD and TDA-only groups, the PPPD group showed 20% better DFS and OS than both the TDA + LND or TDA-only groups (DFS, PPPD vs. TDA + LND, 84.8% vs. 68.8%, *p* = 0.241, PPPD vs. TDA-only, 84.8% vs. 64.1%, *p* = 0.046; OS, PPPD vs. TDA+LND, 89.1% vs. 69.2%, *p* = 0.138, PPPD vs. TDA-only, 89.1% vs. 67.1%, *p* = 0.138). Notably, the performance of LND did not significantly affect DFS or OS in the TDA group, probably due to the non-standardized extent of LND, which currently varies by surgeon and institution. Some studies have recommended clearing the lymph nodes on the supra-duodenal (LN#5) and anterior and posterior portions of the pancreas head (LN#17, LN#13) as the extent of LND during TDA [28,29], but another study did not recommend performing LND at all [18]. Liu F et al. suggested converting to PPPD if regional lymph node sampling during TDA produced positive frozen sections [30]. In addition, an appropriate number of LNs should be taken during oncologic surgery. In this study, many more LNs were harvested from the PPPD group than from the TDA+LND group (PPPD vs. TDA+LND, 15.6 ± 10.5 vs. 4.7 ± 4.4, *p* < 0.001), suggesting that the LND in the TDA procedure was insufficient. Given that N1 stage disease was the most important prognostic factor affecting both DFS and OS in all T stages (Table 3, Table 4), as reported previously [31], TDA could negatively affect recurrence and survival. LN metastasis is high in T1 disease; it is the most significant prognostic factor affecting survival, and distinguishing T1 disease from T2 is difficult both before and during surgery. Therefore, PPPD should be performed when treating T1 disease.

In patients with Tis, the direct comparison of survival was difficult due to the relatively small number of patients; however, as the survival difference in T1 patients became balanced in an analysis of Tis + T1 patients, we conclude that TDA showed an equivalent outcome to PPPD in Tis patients.

Our results take an intermediate position compared with those of other studies. Yoon et al. evaluated the clinical and histopathological results of 201 patients who underwent PD for AoV cancer (67 patients with Tis and T1 disease) and reported that PD should be preferred even for early AoV cancer due to the high incidence of risk factors: lymph node metastasis, PNI, LVI, and CBD or pancreatic duct mucosal involvement (at least one risk factor: 22 patients, 32.8%). They insisted that TDA should be reserved for patients with pTis or pT1 cancer of 1.0 cm or less and high operative risk [25]. Gao et al. compared the surgical and oncological outcomes of 22 TDA patients and 21 PPPD patients and suggested more extended indications for TDA. They performed TDA-only in patients with pTis and pT1 stage disease, tumor size ≤ 2 cm, and an absence of lymph node metastasis. In their T1 patient group without lymph node metastasis, OS did not differ statistically between the TDA and PPPD groups (*p* = 0.927), and the TDA group showed lower surgical morbidity (*p* = 0.033) and lower estimated blood loss (*p* = 0.002). Therefore, they suggested pTis or pT1 stage, tumor size ≤ 2 cm, and negative lymph node metastasis as indications for TDA [17].

Several recent studies have reported that AoV cancer has different prognoses according to the histologic subtypes known as intestinal and pancreatobiliary. The intestinal type of AoV cancer generally shows a better prognosis than the pancreatobiliary type of AoV cancer [32,33]. If the histologic type can be predicted in preoperative imaging studies or endoscopy findings, it could be helpful in selecting appropriate targets for TDA. Chung et al. [34] evaluated the relationship between preoperative magnetic resonance imaging (MRI) or endoscopic findings and pathologic results in AoV cancer. They suggested that an oval filling defect at the distal end of the bile duct on MRI and an extramural protruding appearance with a papillary surface in an endoscopy are likely to suggest the intestinal type of AoV cancer. However, the histologic subtype is not widely used clinically to predict the prognosis of AoV cancer. In this study, only a few cases were reported with pathologic subtypes.

This study has several limitations. First, as a multicenter, retrospective study, the surgical indication and selection criteria for TDA, the preoperative pathologic diagnosis, the surgical technique for TDA, the proper indication for LND in TDA, and the exact locations of the harvested LNs in TDA were not standardized, and the surgeons’ experience in performing TDA was heterogeneous. In addition, the surgeons would have chosen PPPD rather than TDA for more advanced tumors. As a result, most of the clinicopathologic characteristics of the PPPD group indicated more advanced disease than found in the TDA group. These factors obviously produced selection bias. We additionally conducted a PSM analysis to reduce that bias. However, even the PSM analysis failed to offset the differences between the two groups due to much more severe disease pool of the PPPD group among all patients. Second, the number of cases in the TDA group is still too small, even though data were collected from multiple institutions and the sample was larger than in previously reported studies. In particular, only 31 of the TDA patients had T1 stage disease, compared with 188 cases in the PPPD group, which could have affected our statistical analysis as a confounding factor. Third, although the national tumor registration system has the advantage of including a larger number of cases than found in single-center studies, there are limitations to the depth of data. For example, the KOTUS-BP system contains no information about tumor markers and does not include information about the exact cause of death, so we had to compare OS instead of disease-specific survival. Fourth, our study period was too long, about 20 years (from January 2000 to September 2019), and the T and N staging do not reflect the AJCC 8th edition. Because TDA is not a commonly performed surgery, we had to collect all cases entered into the KOTUS-BP system since January 2000. Furthermore, all pathology results in the KOTUS-BP system are based on the 7th edition of the AJCC, so the new 8th edition of the AJCC is not reflected in our results.

## 5. Conclusions

In conclusion, PPPD should be the standard treatment procedure for early AoV cancer. TDA can be applied in patients who have undergone endoscopic papillectomy and been found to have an adenoma with residual tumor, who were diagnosed with a small cancer, or who have many comorbidities, suggestions similar to those in other studies. In addition, because the LN metastasis ratio was about 10% in T1 disease, LND should be added when conducting a TDA. Well-designed studies using a standardized TDA procedure, including the proper extent of LND, are needed to clarify the oncologic safety of using TDA to treat early AoV cancer.

## Figures and Tables

**Figure 1 cancers-13-02038-f001:**
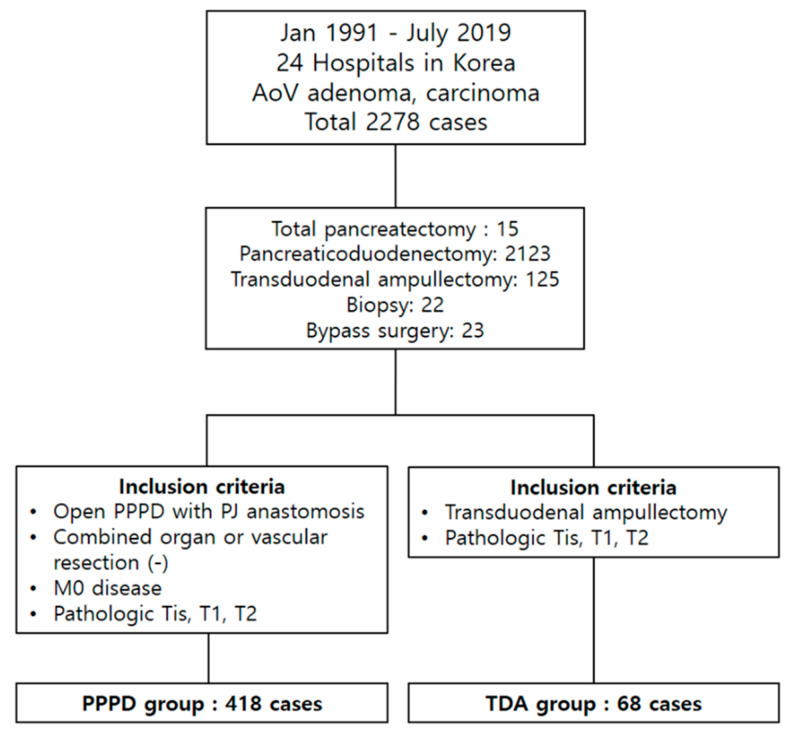
Patient allocation between the PPPD and TDA groups.

**Figure 2 cancers-13-02038-f002:**
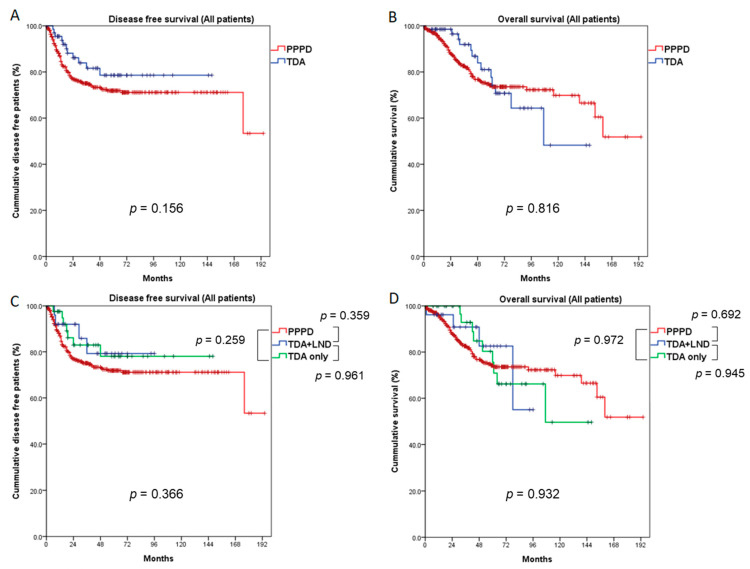
Comparison of survival outcomes between the PPPD and TDA groups (all patients: Tis, T1, and T2). Five-year disease-free survival (**A**) and overall survival (**B**) did not differ between the PPPD and TDA groups. When the TDA group was divided into TDA + LND and TDA-only groups, 5-year disease-free survival (**C**) and overall survival (**D**) did not differ among the three groups. LND: lymph node dissection.

**Figure 3 cancers-13-02038-f003:**
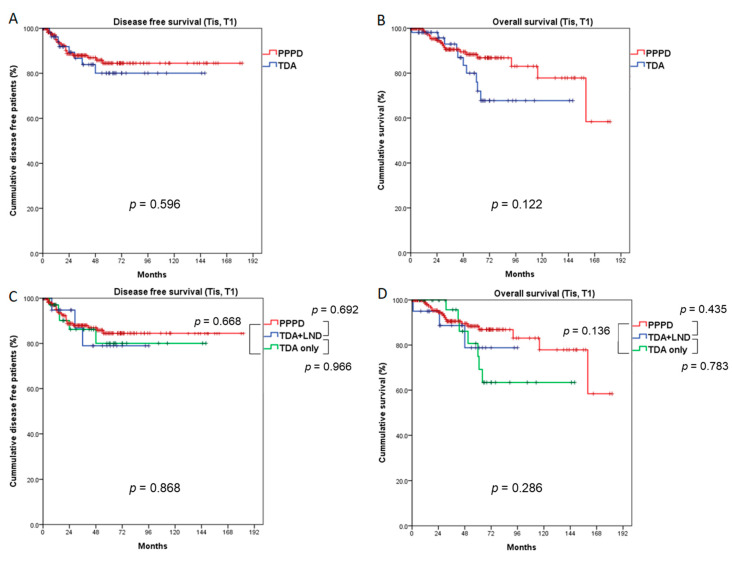
Comparison of survival outcomes between the PPPD and TDA groups (Tis and T1 patients). Five-year disease-free survival (**A**) and overall survival (**B**) did not differ between the PPPD and TDA groups. When the TDA group was divided into TDA + LND and TDA-only groups, the 5-year disease-free survival (**C**) and overall survival (**D**) did not differ among the three groups. LND: lymph node dissection.

**Figure 4 cancers-13-02038-f004:**
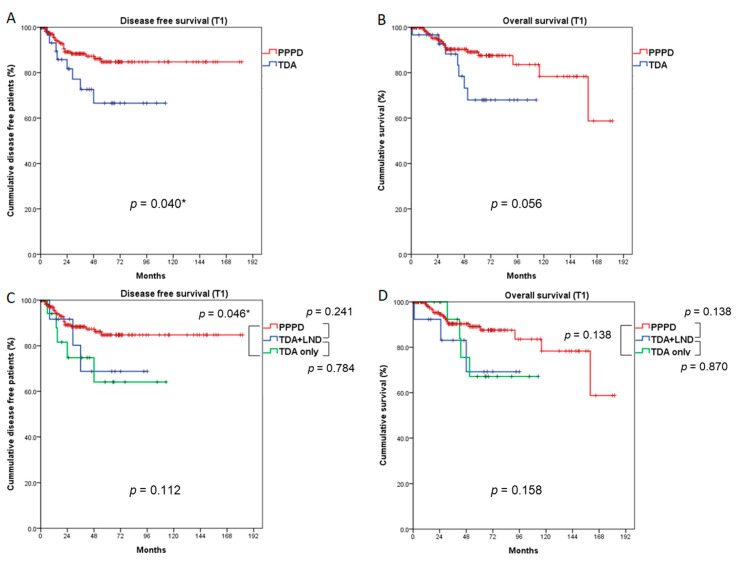
Comparison of survival outcomes between the PPPD and TDA groups (T1 patients). Five-year disease-free survival (**A**) was significantly better after PPPD than TDA (*p* = 0.040). Five-year overall survival (**B**) was marginally better in the PPPD group than the TDA group (*p* = 0.056). When the TDA group was divided into TDA + LND and TDA-only groups, the PPPD group showed better disease-free survival than the TDA-only group (*p* = 0.046), but disease-free survival did not differ between the PPPD and TDA+LND groups (**C**). Five-year overall survival (**D**) did not differ among the three groups. LND: lymph node dissection. ***** statistically significant (*p* < 0.050)

**Table 1 cancers-13-02038-t001:** Comparison of clinicopathologic characteristics and recurrence between the PPPD and TDA groups.

	Tis, T1, and T2			Tis and T1			T1		
	PPPD (418)	TDA (68)	*p*-Value	PPPD (194)	TDA (55)	*p*-Value	PPPD (188)	TDA (31)	*p*-Value
Age (years)	62.5 ± 9.5	61.2 ± 12.7	0.405	63.2 ± 9.1	60.1 ± 12.9	0.105	63.1 ± 9.0	60.6 ± 14.3	0.348
Male (%)	213 (51.0%)	36 (52.9%)	0.761	89 (45.9%)	28 (50.9%)	0.509	86(45.7%)	15(48.4%)	0.785
Size (cm)	1.9 ± 1.0	1.6 ± 0.8	0.034	1.8 ± 1.1	1.6 ± 0.8	0.249	1.8 ± 1.1	1.44 ± 0.8	0.076
T staging			<0.001			<0.001			
Tis	6 (1.4%)	24 (35.3%)		6 (3.1%)	24 (43.6%)				
T1	188 (45.0%)	31 (45.6%)		188 (96.9%)	31 (56.4%)				
T2	224 (53.6%)	13 (19.1%)							
N staging			<0.001			<0.001			<0.001
Nx	10 (2.4%)	42 (61.8%)		6 (3.1%)	35 (63.6%)		6 (3.2%)	18 (58.1%)	
N0	303 (72.5%)	24 (35.3%)		167 (86.1%)	19 (34.5%)		161 (85.6%)	12(38.7%)	
N1	105 (25.1%)	2 (2.9%)		21 (10.8%)	1 (1.85)		21(11.2%)	1(3.2%)	
Differentiation			<0.001			<0.001			0.023
Well	184 (45.0%)	28 (54.9%)		114 (61.0%)	24 (58.5%)		113(62.1%)	22(78.6%)	
Moderate	189 (46.2%)	8 (15.7%)		61 (32.6%)	4 (9.8%)		61(33.5%)	4(14.3%)	
Poorly	29 (7.1%)	1 (2.0%)		6 (3.2%)	0 (0.0%)		6(3.3%)	0	
Undiff.	2 (0.5%)	0 (0.0%)		1 (0.5%)	0 (0.0%)		0	0	
Etc.	5 (1.2%)	14 (27.5%)		5 (2.7%)	13 (31.7%)		2(1.1%)	2(7.1%)	
LVI (+)	128 (36.2%)	0 (0.0%)	0.002	35 (22.6%)	0 (0.0%)	0.077	35(22.9%)	0	0.123
PNI (+)	71 (20.2%)	1 (4.8%)	0.092	22 (14.3%)	0 (0.0%)	0.230	22(14.4%)	0	0.220
Adj. Tx	104 (24.9%)	7 (10.3%)	0.008	18 (9.3%)	1 (1.8%)	0.084	18(9.6%)	1(3.2%)	0.487
R status			<0.001			0.010			0.019
R0	417 (99.8%)	62 (91.2%)		194 (100%)	52 (94.5%)		188(100%)	29(93.5%)	
R1	1 (0.2%)	6 (8.8%)		0	3 (5.5%)		0	2(6.5%)	
Recurrence	93 (22.2%)	11 (16.2%)	0.258	21(10.8%)	8 (14.5%)	0.448	20 (10.6%)	8 (25.8%)	0.036
Recurrence pattern			0.012			0.033			0.015
Local	23 (5.5%)	7 (10.3%)		2 (1.0%)	4 (7.3%)		1 (0.5%)	4 (12.9%)	
Systemic	70 (16.7%)	4 (5.9%)		19 (9.8%)	4 (7.3%)		19 (10.1%)	4 (12.9%)	
Peritoneal seeding	7 (1.7%)	1 (1.5%)	1.000						

PPPD: pylorus-preserving pancreaticoduodenectomy, TDA: transduodenal ampullectomy, Tis: carcinoma in situ, Undiff: undifferentiated, LVI: lymphovascular invasion, PNI: perineural invasion, adj. Tx: adjuvant treatment.

**Table 2 cancers-13-02038-t002:** Comparison of clinicopathologic characteristics and recurrence according to the operation type in patients with T1 stage disease.

	Operation			*p*-Value			
	1. PPPD (188)	2. TDA + LND (13)	3. TDA-Only (18)	1 vs. 2	1 vs. 3	2 vs. 3	Total
Age (years)	63.07 ± 9.03	62.15 ± 11.66	59.39 ± 16.17	0.729	0.354	0.584	0.319
Male (%)	86 (45.7%)	6 (46.2%)	9 (50.0%)	0.977	0.729	0.833	0.942
Size (cm)	1.81 ± 1.09	1.76 ± 0.68	1.20 ± 0.79	0.866	0.025	0.049	0.073
N staging							
Nx	6 (3.2%)		18 (100.0%)	1.000	<0.001	<0.001	<0.001
N0	161 (85.6%)	12 (92.3%)					
N1	21 (11.2%)	1 (7.7%)					
Differentiation							
Well	113 (62.1%)	8 (72.7%)	14 (82.4%)	0.845	0.008	0.253	0.017
Moderate	61 (33.5%)	3 (27.3%)	1 (5.9%)				
Poorly	6 (3.3%)	0 (0.0%)	0 (0.0%)				
Undiff.	0 (0.0%)	0 (0.0%)	0 (0.0%)				
Etc.	2 (1.1%)	0 (0.0%)	2 (11.8%)				
LVI (+)	35 (22.9%)	0 (0.0%)	0 (0.0%)	0.349	0.576	−	0.202
PNI (+)	22 (14.4%)	0 (0.0%)	0 (0.0%)	0.595	1.000	−	0.340
Adj. Tx	18 (9.6%)	1 (7.7%)	0 (0.0%)	1.000	0.377	0.419	0.383
R status							
R0	188 (100.0%)	11 (84.6%)	18 (100.0%)	0.004	−	0.168	<0.001
R1	0 (0.0%)	2 (15.4%)	0 (0.0%)				
Recurrence	20 (10.6%)	3 (23.1%)	5 (27.8%)	0.174	0.050	1.000	0.019
Recurrence pattern				0.249	0.016	1.000	0.004
Local	1 (0.5%)	1 (7.7%)	3 (16.7%)				
Systemic	19 (10.1%)	2 (15.4%)	2 (11.1%)				

PPPD: pylorus preserving pancreaticoduodenectomy, TDA: transduodenal ampullectomy, Undiff.: undifferentiated, LVI: lymphovascular invasion, PNI: perineural invasion, adj. Tx: adjuvant treatment.

**Table 3 cancers-13-02038-t003:** Risk factor analysis for 5-year disease-free and overall survival in all patients.

	Disease Free Survival						Overall Survival					
	Univariate			Multivariate			Univariate			Multivariate		
	OR	95% CI	*p*	OR	95% CI	*p*	OR	95% CI	*p*	OR	95% CI	*p*
Age ≥65 years	1.502	1.024–2.203	0.038				1.976	1.317–2.967	<0.001			
Male	1.132	0.772–1.659	0.525				1.336	0.888–2.012	0.165			
Size ≥1.0 cm	1.481	0.748–2.933	0.260				1.828	0.799–4.183	0.153			
Op method												
PPPD												
TDA + LND	0.630	0.232–1.713	0.365				0.826	0.302–2.256	0.709			
TDA-only	0.646	0.300–1.393	0.265				0.990	0.497–1.973	0.978			
T stage												
Tis												
T1	3.892	0.530–28.611	0.182				0.827	0.287–2.377	0.724			
T2	10.869	1.512–78.153	0.018	827.4	0–∞	0.899	2.115	0.770–5.807	0.146			
N stage												
N0												
Nx	1.195	0.605–2.359	0.608				1.855	0.967–3.559	0.063			
N1	3.841	2.566–5.751	<0.001	2.030	1.263–3.262	0.003	5.396	3.483–8.361	<0.001	3.204	1.806–5.062	<0.001
Differentiation												
Well												
Mod.	2.574	1.643–4.032	<0.001	1.778	1.023–3.089	0.041	2.750	1.701–4.445	<0.001	2.397	1.309–4.388	0.005
Poorly	4.002	2.065–7.755	<0.001	2.059	0.974–4.354	0.059	3.523	1.632–7.605	<0.001	2.343	0.991–5.540	0.052
LVI+	2.580	1.709–3.895	<0.001	1.394	0.865–2.246	0.172	3.449	2.179–5.460	<0.001	1.727	1.026–2.908	0.040
PNI+	1.621	1.028–2.555	0.038	1.206	0.752–1.935	0.437	1.442	0.854–2.435	0.170			

Op: operation, PPPD: pylorus preserving pancreaticoduodenectomy, TDA: transduodenal ampullectomy, LND: lymph node dissection, LVI: lymphovascular invasion, PNI: perineural invasion.

**Table 4 cancers-13-02038-t004:** Risk factor analysis for 5-year disease-free and overall survival in patients with T1 stage disease.

	Disease Free Survival						Overall Survival					
	Univariate			Multivariate			Univariate			Multivariate		
	OR	95% CI	*p*	OR	95% CI	*p*	OR	95% CI	*p*	OR	95% CI	*p*
Age ≥65	1.696	0.807–3.567	0.163				2.249	1.008–5.018	0.048			
Male	0.537	0.243–1.187	0.124				1.021	0.466–2.238	0.959			
Size ≥1.0 cm	1.315	0.456–3.789	0.613				2.047	0.479–8.751	0.334			
Op method												
PPPD												
TDA + LND	2.033	0.604–6.844	0.252				2.473	0.720–8.493	0.150			
TDA-only	2.500	0.938–6.664	0.067				2.211	0.739–6.617	0.156			
N stage												
N0												
Nx	3.045	1.157–8.014	0.024	−	−	−	4.228	1.56611.413	0.004	2.863	0.526–15.584	0.224
N1	7.920	3.362–18.655	<0.001	2.148	0.645–7.146	0.213	13.214	5.157–33.859	<0.001	3.898	1.024–14.849	0.046
Differentiation												
Well												
Mod.	3.035	1.278–7.204	0.012	1.924	0.576–6.425	0.287	1.313	0.543–3.178	0.545			
Poorly	11.104	2.948–41.821	<0.001	5.446	1.073–27.650	0.041	8.613	1.829–40.557	0.006	1.303	0.222–7.641	0.770
LVI+	4.108	1.617–10.434	0.003	1.725	0.546–5.446	0.353	11.960	3.729–38.362	<0.001	6.933	1.594–30.160	0.010
PNI+	4.124	1.683–10.107	0.002	2.953	1.016–8.578	0.047	2.613	0.803–8.502	0.111			

Op: operation, PPPD: pylorus preserving pancreaticoduodenectomy, TDA: transduodenal ampullectomy, LND: lymph node dissection, LVI: lymphovascular invasion, PNI: perineural invasion.

## Data Availability

Data are available in a publicly accessible repository.

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
