# Peer review of "Comparison of Oncologic Outcomes between Transduodenal Ampullectomy and Pancreatoduodenectomy in Ampulla of Vater Cancer: Korean Multicenter Study"

_cancers, 2021, doi:10.3390/cancers13092038_

Round 1
Reviewer 1 Report
All the comments are appropriately responded. Especially, the revised conclusion is acceptable.
Reviewer 2 Report
This revised manuscript has been corrected well.
This manuscript is a resubmission of an earlier submission. The following is a list of the peer review reports and author responses from that submission.
Round 1
Reviewer 1 Report
Comparison of oncologic outcomes between transduodenal ampullectomy and pancreatoduodenectomy in ampulla of Vater cancer: Korean multicenter study
Comments to the authors
 Transduodenal ampullectomy (TDA) is less invasive surgical procedure, but it has some clinical concerns, especially about oncological outcome. In this study, authors stated that TDA did not compromise oncologic outcomes in early ampulla of Vater (AoV) cancer compared to pylorus-preserving pancreatoduodenectomy (PPPD). Reviewer has some comments to the authors. Firstly, authors just list the results even in discussion and conclusion. Authors should clearly claim whether TDA is appropriate procedure for the radical treatment of early AoV cancer. Secondly, authors referred that no difference of survival outcomes between the PPPD and TDA in conclusions. This study showed no difference of survival outcomes between PPPD and TAD in Tis+T1 Stage, but TDA had worse results in T1 Stage compared to PPPD. This suggests that TDA is applicable to Tis Stage only. Also, it is necessary to mention that clinicopathologic characteristics are worse in the PPPD group. Thirdly, there is a lack of information about the indication of lymph node dissection. Lesions or harvested numbers lymph node dissection are important information when considering lymph node metastasis, recurrence and clinical outcomes. Moreover, strong selection bias for lymph node dissection among surgeons made evidence of this study weak.
Reviewer has other minor comments that;
- It should be stated that PD is standard surgical procedure for early stage.
- Definition of Recurrence pattern is unclear.
- Recent years, several studies reported that AoV cancer have different prognosis due to subtype (intestinal type and pancreatobiliary type). Please discuss this in their discussion.
Reviewer 2 Report
Thank you for the opportunity to review this paper. Authors presented multicenter study of ampulla of Vater (AoV) cancer to compare oncological safety of transduodenal ampullectomy (TDA) with that of pylorus-preserving pancreatoduodenectomy (PPPD). They concluded that, compared with PPPD, TDA with LND did not compromise oncologic outcomes in early AoV cancer.I believe this article contained important information. However, the followings are the list of my comments which should be addressed.
Comments
- The most serious limitation of this study was the selection bias of the patients. In other words, how did surgeons decide the procedure, PPPD or TDA? To improve upon this problem, propensity score matched analysis should be done.
- Author said that the oncologic behavior in the PPPD group showed more aggressive than that in the TDA group. I think that authors simply chose PPPD than TDA for advanced tumor. So, this description will mislead readers.
- Tumor markers should be stated and compared.
- In figure 2 A and B, DFS was better in TDA and OS was better in PPPD group (although difference was not significant). Death from other disease was more often TDA? Disease specific survival should be shown.
- Even though the difference was not significant, DFS (84.8% vs. 68.8%) and OS (89.1% vs. 69.2%) was worse in TDA. So, I can’t agree with the current conclusion.
- Dissemination of cancer cell during surgery was one of the risks of TDA. Recurrence pattern should be stated in detail. Especially, the incidence of peritoneal metastasis is important.